# *Staphylococcus aureus* Behavior on Artificial Surfaces Mimicking Bone Environment

**DOI:** 10.3390/pathogens12030384

**Published:** 2023-02-28

**Authors:** Anaïs Lemaire, Jennifer Varin-Simon, Fabien Lamret, Marie Dubus, Halima Kerdjoudj, Frédéric Velard, Sophie C. Gangloff, Fany Reffuveille

**Affiliations:** 1Biomatériaux et Inflammation en Site Osseux, Université de Reims Champagne-Ardenne, BIOS EA 4691, SFR Cap Santé, 51097 Reims, France; 2UFR d’Odontologie, Université de Reims Champagne-Ardenne, 51097 Reims, France; 3Service de Microbiologie, UFR de Pharmacie, Université de Reims Champagne-Ardenne, 51097 Reims, France

**Keywords:** *Staphylococcus aureus*, adhesion, biofilm, bone substitutes

## Abstract

Infections, which interfere with bone regeneration, may be a critical issue to consider during the development of biomimetic material. Calcium phosphate (CaP) and type I collagen substrates, both suitable for bone-regeneration dedicated scaffolds, may favor bacterial adhesion. *Staphylococcus aureus* possesses adhesins that allow binding to CaP or collagen. After their adhesion, bacteria may develop structures highly tolerant to immune system attacks or antibiotic treatments: the biofilms. Thus, the choice of material used for scaffolds intended for bone sites is essential to provide devices with the ability to prevent bone and joint infections by limiting bacterial adhesion. In this study, we compared the adhesion of three different *S. aureus* strains (CIP 53.154, SH1000, and USA300) on collagen- and CaP-coating. Our objective was to evaluate the capacity of bacteria to adhere to these different bone-mimicking coated supports to better control the risk of infection. The three strains were able to adhere to CaP and collagen. The visible matrix components were more important on CaP- than on collagen-coating. However, this difference was not reflected in biofilm gene expression for which no change was observed between the two tested surfaces. Another objective was to evaluate these bone-mimicking coatings for the development of an in vitro model. Thus, CaP, collagen-coatings, and the titanium-mimicking prosthesis were simultaneously tested in the same bacterial culture. No significant differences were found compared to adhesion on surfaces independently tested. In conclusion, these coatings used as bone substitutes can easily be colonized by bacteria, especially CaP-coating, and must be used with an addition of antimicrobial molecules or strategies to avoid bacterial biofilm development.

## 1. Introduction

A bone defect is a common health problem due to trauma, infection, tumor, or congenital disease [1]. The synthetic bone substitute may be used to ensure healing. The interest in the development of biomimetic material in bone tissue engineering is increasing [2] to repair bone defects and restore bone integrity [3]. However, contamination during surgery, from a hematogenous way or from an infected adjacent tissue could lead to infection on the bone site [4]. In the case of total joint arthroplasty, the incidence of infection ranges from 1% or less in primaries to 5% in aseptic revision settings and 20% or more when revising for infection [5]. This can lead to dramatic consequences (irreversible sequelae, chronic infection, and death). Furthermore, the overuse of antibiotics during the treatment of such infections leads to an antibiotic-resistance rise, especially if the infection turns out to be chronic [6]. The persistence of the infection can start with the adhesion of bacteria on bone or material, which can promote biofilm formation. This heterogeneous bacteria community is highly antibiotic and immune system-tolerant [6]. Recently, scientists have shown that biofilm formation is strongly dependent on its environment and especially on the nature of the surface it grows on [7]. Thus, the development of realistic models is needed to better decipher bacterial biofilm formation [7]. In the perspective of biomaterial development, testing biofilm formation on bone-related materials is also essential.

The mineral phase of bone is mainly composed of calcium phosphate (CaP), whereas the organic phase is mainly composed of type I collagen [8]. Both could be used for biomaterial development or as bone-mimicking surfaces in an in vitro model. On the one hand, hydroxyapatite (HA), belonging to the CaP family has been used for its inorganic composition similar to natural bone [1]. On the other hand, many collagen biomaterials are cell-adhesive and support cell differentiation, favoring bone repair [9]. However, CaP or collagen also favor bacterial adhesion. For example, *S. aureus*, the main species responsible for bone and joint infections [6], is able to adhere to calcium phosphate- and collagen-coatings thanks to adhesin such as Cna, a collagen adhesin [10].

In this study, we compared the adhesion of three different *S. aureus* strains on different bone-mimicking surfaces: collagen- and CaP-coatings. Our objective was to compare the bacterial behavior on each surface in the same experimental approach to select the coating presenting the lowest risk of bacterial colonization. Thus, we evaluated and compared the bacterial adhesion rate, the structure and composition of the biofilms formed on the coatings, and the expression of genes known to be involved in bacterial adhesion or biofilm development. We also evaluated *S. aureus* adhesion on these coated supports in the simultaneous presence of a fibronectin-coated titanium surface to determine if the presence of various types of surfaces at the same time influenced bacterial behavior. Indeed, for future development of an in vitro model of bone and joint infection, different materials mimicking bone surfaces should be present to be closer to in vivo conditions where bacteria could adhere to the bone matrix or on titanium prosthesis.

## 2. Materials and Methods

### 2.1. Coating Preparation

Before the coating step, glass coverslips (12 mm diameter) were immerged 10 min in acetone, 4 times 5 min in 95% of alcohol, 2 times 5 min in 100% alcohol, and dried and sterilized at 180 °C for 2 h. The preparation of collagen-coating was processed using glass coverslips which were immersed in 50 mL tube containing rat collagen solution at 0.381 mg/mL (BD Biosciences, CA, USA) and incubated overnight at 37 °C. Coverslips were then deposited in 24-well plates to be washed and dried, before being processed the same day.

Glass coverslips coated with calcium phosphates (14 mm diameter, Thermo Fisher Scientific, Waltham, MA, USA) were prepared according to [Dubus, 2020]. Powders of calcium nitrate (Ca(NO_3_)_2_•4H_2_O), diammonium hydrogen phosphate ((NH_4_)_2_HPO_4_), and Tris(hydroxymethyl) aminomethane (Tris) from Lifecore Biomedical were used. The salt solutions were prepared in ultrapure water (Millipore, Guyancourt, France). A calcium solution of Ca(NO_3_)_2_•4H_2_O (0.32 M) and a phosphate solution of (NH_4_)_2_HPO_4_ (0.2 M) were prepared in Tris buffer (10 mM Tris, pH 4 and 10, respectively), as described previously [11]. Before each experiment, a cleaning step of the coverslips with sodium dodecyl sulphate (100 mM, Sigma-Aldrich, Saint-Louis, MO, USA) was performed for 15 min at 100 °C. After an intensive rinse with ultrapure water, the coverslips were placed in contact with HCl (100 mM, Sigma-Aldrich, Saint-Louis, MO, USA) for 15 min at 100 °C, rinsed with ultrapure water, and stored at 4 °C. For the constitution of CaP, an automated spraying device was used. This device is assembled from four identical Airbrushes VL (Paasche, WI, USA). Each nozzle is pressurized by an internal compressed air line at a pressure of 1 bar and connected to solenoid valves. The spraying of the different solutions, according to a chosen deposition sequence, is carried out by a succession of valve closures and openings controlled by in-house software. Three nozzles are used to spray calcium, phosphate, and rinsing solutions, respectively. The fourth nozzle, free of the solution, is dedicated to the drying stage. The support is placed vertically on a mobile support. For a homogeneous accumulation of the CaP layers, the support is rotated at 150 rpm. For the formation of the coating, the calcium and phosphate solutions were sprayed simultaneously for 2 s, followed by a 2-s rinsing step with ultrapure water, and a 2-s drying step under compressed air.

Titanium disks (diameter 12 mm × height 5 mm) (ACNIS, Chassieu, France) were shotpeened by CRITT-MDTS (Charleville-Mézières, France). Titanium disk surfaces were coated with fibronectin (Sigma-Aldrich, Saint-Louis, MO, USA) at a concentration of 1 mg/mL. Two hundred microliters of surface tension drop forming solution was deposited on the surface of titanium disks before a 2 h incubation at room temperature. After washing, the disks were dried overnight under the laminar flow.

### 2.2. Bacterial Strains and Culture Media

Three strains of *S. aureus* were used in this study, namely, a methicillin-resistant strain USA300 isolated in the 1990s in the USA and two methicillin-susceptible strains: CIP 53.154 (ATCC 9144) isolated in Oxford in 1944, used as referent strain in European Pharmacopeia; and SH1000, derivate from the strain 8325-4, with the rsbU gene reconstituted, which forms strong biofilms. Trypto-Casein Soy Agar (TSA, Biokar, Allonne, France) was used for the isolation and cultivation of tested bacteria.

For all experiments, the absorbance of overnight cultures (18 h of culture) in nutrient broth (BioRad, Hercules, San Jose, CA, USA) was adjusted to 1 at 600 nm, before being diluted 1:100 in minimal medium (MM). MM is composed of 62 mM potassium phosphate buffer, pH 7.0, 7 mM (NH_4_)_2_SO_4_, 2 mM MgSO_4_, and 10 µM FeSO_4_ containing 0.4% (*w*/*v*) glucose and 0.1% (*w*/*v*) casamino acids [12].

### 2.3. Numeration of Viable Adherent Bacteria

To form an initial biofilm, 500 µL of bacterial suspension was inoculated in a 24-well plate and incubated at 37 °C, for 24 h, under anaerobic atmosphere using the GenBag system (Biomérieux, Marcy-l’Étoile, France) containing different coated supports. After 24 h of incubation at 37 °C under anaerobic conditions using the GenBox system (Biomerieux, Marcy l’Etoile, France), the coated supports were washed to remove non-adherent bacteria and supports were then transferred to a 15 mL tube containing 2 mL of MM. Adherent bacteria were then detached by an ultrasonic bath (40 kHz) for 5 min. Bacteria before and after ultrasound were plated on TCS agar plates using automatic spiral (Easy Spiral, Interscience, Saint-Nom-la-Bretèche, France). After 24 h of incubation, the number of colony-forming units (CFU) was determined using automatic counter SCAN (Interscience, Saint-Nom-la-Bretèche, France), and the quantity of adherent bacteria was calculated. Six independent replicates of the experiment were performed.

For competitive adhesion, the experiments were performed in a 6-well plate. Six milliliters of bacterial suspension in MM was deposited per well, containing simultaneously a collagen-coating, a CaP-coating, and a fibronectin-coating. After 24 h of incubation, under the anaerobic atmosphere, with supports, at 37 °C, the number of living adherent bacteria was determined as previously detailed. To determine the number of bacteria per mm^2^, results were normalized by the surface area of each support. The experiment was performed on 6 independent replicates.

### 2.4. Scanning Electron Microscopy (SEM)

Adhered bacteria on supports, in 24-well plates, 24 h, at 37 °C and under an anaerobic atmosphere, were washed twice with phosphate buffered saline (PBS) for 5 min, then fixed with 2.5% (*w/v*) glutaraldehyde for 1 h (Sigma-Aldrich, Saint-Louis, MO, USA). The samples were then washed twice with distilled water for 10 min and dehydrated by successive treatment in graduated ethanol solutions (50, 70, 90, and twice 100%) for 10 min. The initial biofilms were finally dried in a drop of hexamethyldisilazane (HMDS) (Sigma, Saint-Louis, MO, USA). After air-drying at room temperature, the samples were sputtered with a thin gold-palladium film using a JEOL JFC 1100 ion sputtering instrument (Tokyo, Japan). The biofilms were observed using a Schottky field emission scanning electron microscope (JEOL JSM-7900F, Tokyo, Japan). Images were obtained at a primary beam energy of 2 kV (SM-EXG65 electron emitter). The experiment was performed 2 times.

### 2.5. Confocal Laser Scanning Microscopy (CLSM)

After 24 h incubation at 37 °C, bacteria biofilms were washed twice in PBS and then stained with (i) SYTO™ 9 at 1 µM and propidium iodide (Thermo Fisher Scientific, Waltham, MA, USA) at 20 µM, labelling, respectively, live and damaged or ‘dead’ bacteria, (ii) SYPRO^®^ Ruby (*v*/*v*) labelling proteins, or (iii) wheat germ agglutinin (WGA) combined with Alexa Fluor™ 350 conjugate at 100 mg/mL to label PIA [13] (iv) with TOTO^TM^-3 (Thermo Fisher Scientific, Waltham, MA, USA) at 2 µM labelling extracellular DNA. Each fluorochrome was diluted in 0.9% NaCl. After 30 min of incubation in the dark at room temperature, each slide was washed twice in PBS and placed in a 24-well crystal plate with a glass bottom (Porvair, UK) containing PBS. The biofilm face was turned down for observation by confocal laser scanning microscopy (CLSM) (LSM 710 NLO, Zeiss, Germany). Labelled compounds were imaged, and their volume was quantified using IMARIS software (Imaris, v9.5.1). The experiment was performed on two independent replicates, with 3 captures for each sample. The quantity of each fluorescent marker was normalized by SYTO^TM^ 9 fluorescence volumes to obtain a percentage.

### 2.6. RT-qPCR (RNA Purification and Reverse Transcription)

Bacteria from planktonic growth or adherent were rapidly collected by centrifugation at 5000 rpm, 5 min, at room temperature. Pellets were directly stocked at −80 °C for long-term conservation. Total RNAs were extracted using the MasterPure^™^ RNA purification kit (Lucigen, Middleton, WI, USA) according to the manufacturer’s recommendations. The total RNAs were then reverse transcribed into complementary DNA (cDNA) using a high capacity cDNA reverse transcription kit (Applied Biosystems, Waltham, MA, USA) according to the manufacturer’s instructions. Finally, transcripts were amplified by RT-qPCR using SYBR^™^ Green Master Mix (Applied Biosystems, Waltham, MA, USA) and on a StepOne Plus^™^ system (Applied Biosystems, Waltham, MA, USA). A data analysis was performed with StepOne^™^ Software v2.3 (Applied Biosystems, Waltham, MA, USA). Target transcript levels (N-target) were normalized to the housekeeping gene transcript levels and messenger RNA (mRNA) level with the equation N_target_ = 2^−∆Ct^, where δCt was the Ct value of the target gene after subtracting the Ct of the housekeeping gene. *gyrB* was used as the housekeeping gene for CIP 53.154; *rho* was used for SH1000 and USA300. The experiment was performed on 3 independent replicates, for each strain.

### 2.7. Graphical Representation of Data and Statistical Analysis

Data are expressed as histograms representing the mean and error bars representing the standard deviation, and green dots represent independent biological replicates. For confocal microscopy quantifications, different area acquisitions and biological replicates were considered. All statistical analyses were evaluated using the non-parametric Mann–Whitney U exact test for independent samples (GraphPah Prism, 8.0.2). Differences were considered significant at *p* < 0.05.

## 3. Results

### 3.1. Bacterial Adhesion on Coated Supports Mimicking Bone Matrix

First, we studied the bacteria adherence of three *S. aureus* strains (CIP 53.154, SH1000, and USA300) on the glass support coated with collagen or with CaP (Figure 1A). The three strains showed the same adhesion levels: about 10^6^ bacteria/cm^2^ on collagen and from 10^7^ to 10^8^ bacteria/cm^2^ on CaP. These results underlined a highest bacteria adhesion on collagen- or CaP-coatings compared to the adhesion rate on the classical plastic support routinely used for adhesion experiment (between 10^5^ and 10^6^ bacteria/cm^2^ on plastic, Appendix A). SH1000 strains showed the highest but more variable rate of adhesion on CaP-coating. The USA300 adhesion rate was more stable and was significantly higher on CaP-coating than on collagen-coating.

Then, we explored the adhered bacteria aspect by SEM (Figure 1B). Although no matrix was visible for the SH1000 strain, fibrous matrix (red arrows) for CIP 53.154 and pili-*like* structures (blue arrows) for USA300 were observed on glass coated with collagen. A fibrous matrix (red arrows), for all strains, on glass coated with CaP was identified.

### 3.2. Initial Biofilm Structures and Composition on Collagen and CaP Supports

Biofilm matrix composition was characterized by specific fluorescent labels and observed using CLSM (Figure 2). A non-significant increase in the proportion of fluorescent bacteria was observed on the CaP compared to collagen for all strains. However, we noticed a high proportion of dead or damaged bacteria on CaP for the CIP 53.154 strain (Figure 2A).

Following the observation of a larger matrix on CaP by SEM, we were interested in the composition of the biofilm matrix on this support. The biofilm matrix of CIP 53.154 and SH1000 strains was similar with a predominant presence of exopolysaccharides (WGA) and eDNA (TOTO^TM^-3) in the same relative quantity. For the USA300 strain, the relative quantity of the biofilm matrix was inferior compared to the two other strains. The proportion of proteins was strain-dependent, with a more important proportion of proteins for USA300 and less for CIP 53.154 (Figure 2B,C). Finally, bacteria and matrix aggregates were observed in 3D-reconstruction on CaP (Figure 2C). We observed a different structure organization with the presence of aggregates for CIP 53.154 strains with some of them composed of dead bacteria, and irregular monolayers for SH1000 and USA 300 strains. Proteins formed some “clusters” especially for USA300 but were also observed for CIP 53.154. The polysaccharides and eDNA components of the matrix were present everywhere on the sample, in the same proportion. However, TOTO^TM^-3 seemed more present in the CIP 53.154 biofilm.

### 3.3. Initial Biofilm Formation-Related Gene Expression on Collagen and CaP Supports

To test the hypothesis of a differential mRNA expression of bacteria on collagen and CaP, we selected a set of genes already identified as related to biofilm development (Table 1). The stress and biofilm-related genes’ expression was quantified by RT-qPCR. The results obtained from adherent bacteria were normalized to the planktonic bacteria data to observe the up- or downregulation of gene expression in biofilm (Figure 3).

The expression of stress-related genes involved in biofilm initiation was studied: SOS system-related genes *recA* and *lexA*, regulator-related gene *sigB*, and ppGpp-mediated response-related gene *rsh* [14,15]. A non-significant change in the expression was observed, probably due to the high heterogeneity of the bacteria population composing the biofilm (Figure 3A–C), even if the *rsh* gene was still downregulated for the SH1000 strain in biofilm compared to the planktonic state. In addition, the non-significant overexpression of *sigB* on the CaP but not on collagen could be supposed for the CIP 53.154 and USA strains but not the SH1000 one (Figure 3B,C).

Then, the expression of genes related to biofilm formation was quantified: extracellular DNA-related gene *cidA*, biofilm development-related gene *sarA*, adhesion-related gene *fnbpB* and the autoinducer peptide (AIP), and *quorum sensing*-related gene *agrB* selected thanks to the literature review [15,16,17,18]. A non-significant overexpression of *fnbpB* on CaP was observed for the CIP 53.154 strain (Figure 3D,E). The USA300 strain seemed to overexpress *agrB* genes on CaP contrary to the two other strains (Figure 3F).

### 3.4. Influence of the Simultaneous Presence of Different Coated Supports on S. aureus Adhesion

In this approach, the objective was to evaluate *S. aureus* behavior in the presence of different types of surfaces as found in prosthesis infection [19,20], including CaP- and collagen-coating as supports mimicking bone matrix. The experiment was performed with the addition of a third support: fibronectin-coated titanium mimicking the presence of a prosthesis during a bone and prosthesis infection. Thus, in this experiment, we observed bacterial adhesion in the simultaneous presence of different supports: fibronectin-coated shotpeened titanium, collagen-coating, and CaP-coating (Figure 4). We still observed a more important quantity of bacteria on CaP surfaces than the other ones but without significant difference (Figure 4), except for CIP 53.154 with the highest rate.

## 4. Discussion

In this study, we compared the behavior of three *S. aureus* strains on collagen and CaP substrates that were used as bone-mimicking surfaces for the development of biomaterial with the objective of selecting the coating with the lesser risk of infection. Thus, we evaluated the adhesion capacity of bacteria, the nature and structure of biofilms that could reflect the ability of colonization of bacteria, but also the expression of genes that are already known to be involved in biofilm formation. The three strains CIP 53.154, SH1000, and USA300 highly adhered to these surfaces. CaP-sprayed surfaces carried the greatest amount of bacteria per cm^2^. However, the results were heterogeneous according to strain and coating, underlining a high variability of behavior. This experiment also underlined that bacterial behavior varied according to the type of surface and other options than plastic should be considered to evaluate antimicrobial and/or antiadhesive molecules destined to non-plastic biomaterial.

Thanks to SEM images, we observed that bacteria formed a substantial matrix on CaP-coating, confirmed with fluorochromes labelling and confocal microscopy. We speculated that the quantity of the matrix formed on collagen surfaces seemed strain-dependent as the SH1000 was not producing matrix at all. Interestingly, the CIP 53.154 strains demonstrated a high rate of dead or damaged bacteria on the CaP surface, which could explain the origin of the important matrix structure. In a previous publication, the dead bacteria proportion in the CIP 53.154 biofilm was different according to oxygen proportion [21]. Taken all together, we confirmed that the biofilm composition of this strain changed according to its environment [22]. Moreover, some bacteria-forming aggregates looked completely dead, while others were alive, showing a high heterogeneity of bacteria within the initial biofilm, and explaining the difficulty to obtain stable qPCR results and heterogeneity of the genes’ expression as, for example, the *cidA* gene, involved in bacterial lysis. The presence of TOTO^TM^-3 fluorochrome-labelling eDNA did not seem to be linked to the presence of the dead bacteria parts in the initial biofilm. This suggests that earlier lysis events may have occurred allowing for the more uniform presence of extracellular DNA.

The biofilm matrix formed on this support was less important for the USA strain but mainly composed of exopolysaccharides for all strains. However, the proportion of proteins was strain-dependent, confirming the previous description of *S. aureus* biofilm [23].

The environment could induce and strongly influence the biofilm formation [7]. Based on the literature, we compared the expression of genes identified as involved in bacterial adhesion in response to environment conditions. In this study, no important change in biofilm-involved genes was observed except a potentially increased expression of the *rsh* gene during the adhesion to CaP compared to collagen support. Calcium ions could be released from CaP support [11], inducing a stress response and influencing bacteria behavior as the calcium ion is involved in many bacterial metabolism pathways [24,25]. We noticed a non-significant overexpression of two biofilm-involved genes *sarA* and *agrB* [26] on the CaP surface only for the USA300 strain, underlining the potential affinity of USA300 for CaP support to initiate and reinforce biofilm in a different way than the two other tested strains.

Whereas the behavior of *S. aureus* on biomimetic CaP- and collagen-coatings was variable, CaP-coating seemed to favor bacterial adhesion, more than collagen-coating, not only by the quantity of adhered bacteria but also with the production of a thicker biofilm matrix. This matrix structure could facilitate the adhesion of other microorganisms and protect embedded bacteria. Indeed, a previous study underlined that the introduction of a CaP bone graft could lead to the integration of the bacteria into the CaP, delaying the time of infection [5]. Many strategies were developed to combine antimicrobial with bone substitutes [27]. The addition of an antimicrobial molecule such as chitosan in CaP preparation reduced the risk of bacterial contamination [28].

For the perspective of future in vitro model development, the second approach aimed to compare bacteria adhesion behavior when the support was alone or when all bone-mimicking surfaces were simultaneously put together. We still observed the same behavior of adhered bacteria, with the highest quantity on CaP-coating, maybe due to an easier adhesion, a better proliferation, or a higher available area for bacteria because of the presence of thick structures. This underlined that the presence of different types of surfaces, in the same site, did not change the fact that the quantity of bacteria was more important on CaP-coating. Comparing the three strains, the SH1000 strain was the one that adhered the most when CaP-coating was alone in the well, whereas the CIP 53.154 adhered the most to CaP when it was in the presence of fibronectin-coating. This slight behavior modification underlines a possible change due to the tested conditions and highlights the necessity to choose the good parameters to test a biomaterial or to develop an in vitro model.

The use of CaP-coating glass and collagen-coating glass in an in vitro model could be considered for the pre-screening of antibiofilm molecules; however, the lack of very specific 3D architecture, as in bone and its matrix, strongly limits the similarity and therefore the real behavior of bacteria in vivo.

## 5. Conclusions

In conclusion, CaP-based bone substitutes allow *S. aureus* adhesion and could enhance the risk of bone and joint infection. A collagen bone substitute seems to be less risky, but is still easily subject to bacteria adhesion, especially if collagen is used in a 3D structure, which offers more surfaces and the possibility for bacteria to adhere.

## Figures and Tables

**Figure 1 pathogens-12-00384-f001:**
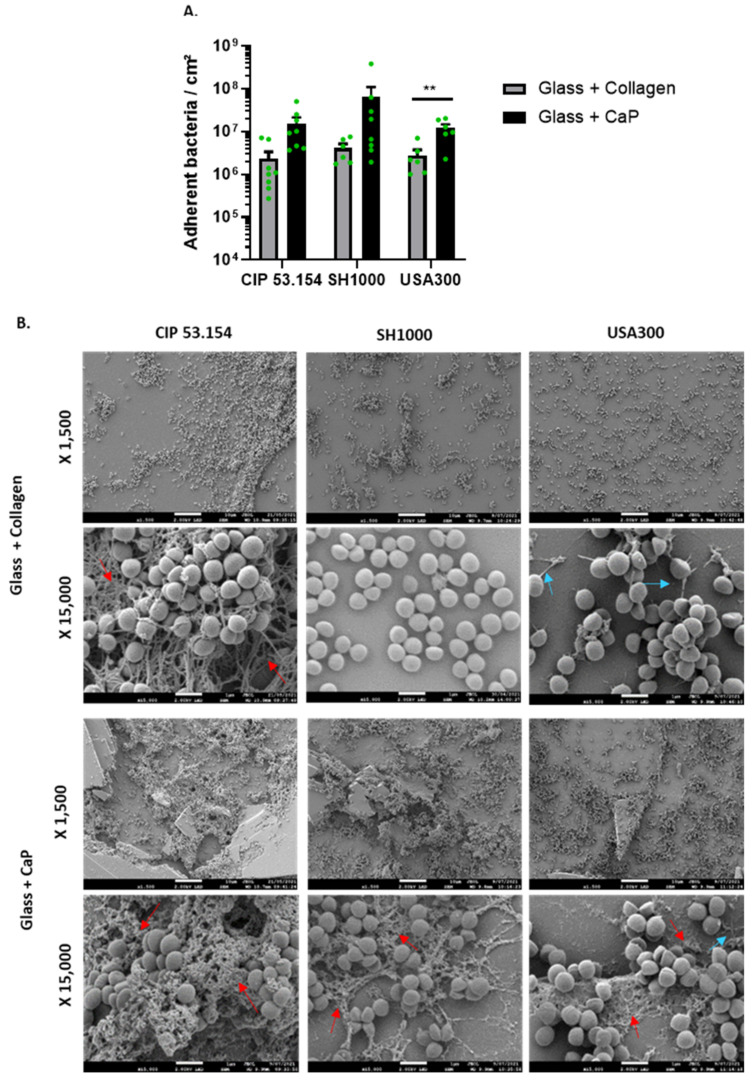
*S. aureus* adhesion on glass coated with collagen or calcium phosphate. (**A**) Results represent the number of viable adherent bacteria per cm^2^ for CIP 53.154, SH1000, and USA300 on different supports, *n* = 4 to 8 (green dots represent independent biological replicates). Wilcoxon–Mann–Whitney test **, *p* < 0.01. (**B**) Representative images of scanning electron microscopy acquisitions (magnification ×1500 and ×15,000). Red arrows show the fibrous matrix; blue arrows show pili-like structures. The scale bars indicate 1 µm, *n* = 2.

**Figure 2 pathogens-12-00384-f002:**
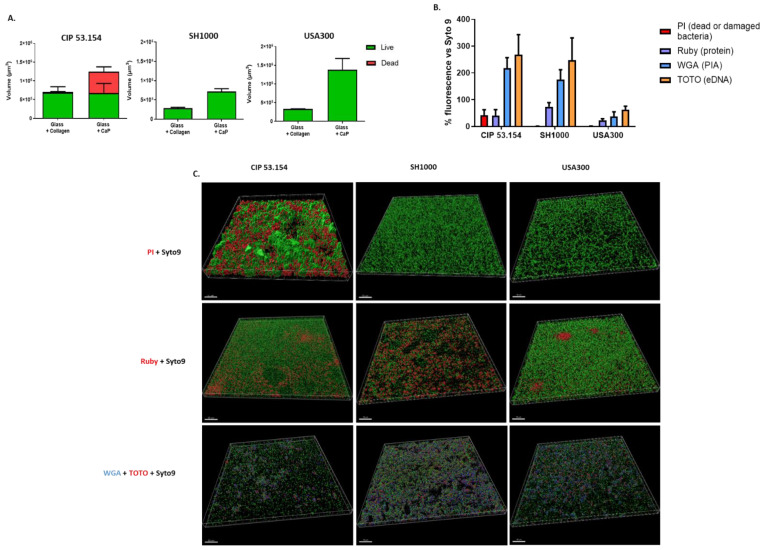
Characterization of S. aureus biofilms on different supports. (**A**) Volumes of fluorescent staining with repartition of SYTO^TM^ 9 and PI staining. The graph shows the fluorescence volumes overlapping on top of each other. (**B**) Percentage of SYPRO^®^ Ruby (purple, protein), WGA (blue, PIA), and TOTO^TM^-3 (orange, extracellular DNA) acquired by confocal microscopy on glass coated in calcium phosphate. (**C**) Representative 3D views of matrix biofilm composition on glass coated in calcium phosphate. Scale bar indicates 50 µm, *n* = 2.

**Figure 3 pathogens-12-00384-f003:**
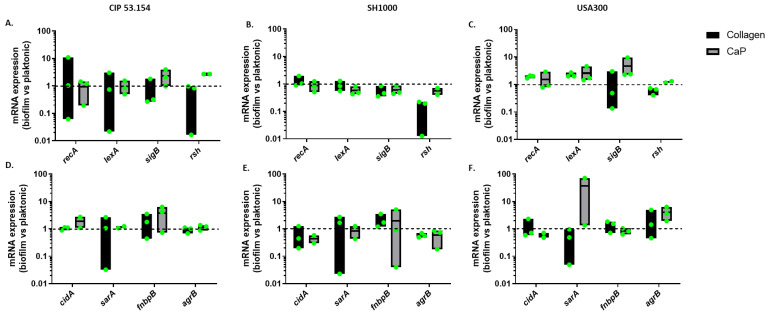
*S. aureus* gene relative expression on different surfaces. Results represent relative mRNA expressions of bacteria within biofilm vs. planktonic bacteria. Expression of stress-related genes for CIP 53.154 (**A**), SH1000 (**B**), and USA300 (**C**). Expression of genes related to biofilm formation for CIP 53.154 (**D**), SH1000 (**E**), and USA300 (**F**). Baseline = 1; *n* = 2 to 3 (green dots represent independent biological replicates).

**Figure 4 pathogens-12-00384-f004:**
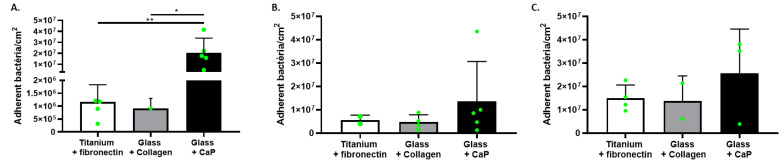
*S. aureus* adhesion in the simultaneous presence of different coatings. Results represent the number of viable adherent bacteria per cm^2^ in presence of both supports: titanium coated with fibronectin, glass coated with collagen, and glass coated with calcium phosphate. (**A**) CIP 53.154, (**B**) SH1000, (**C**) USA300, *n* = 2 to 5 (green dots represent independent biological replicates). Wilcoxon–Mann–Whitney test * *p* < 0.05, ** *p* < 0.01.

**Table 1 pathogens-12-00384-t001:** Nucleotide sequences of primers used for qPCR.

Target Gene	Forward Primer	Reverse Primer
*gyrB*	CACGTGAAGGTATGACAGCA	ACAACTTGACGCACTTCAGA
*rho*	AACGTGGGGATAAAGTAACTGG	TTCACTTCTTCTGCGTTATGGT
*recA*	ATAGGTCGCCGAGTTTCAAC	GCGCTACTGTTGTCTTACCA
*lexA*	TCAATATTTTCTACTGCGGTAATAGG	GAAACGATTCATGTGCCAGTT
*sigB*	TTGTCCCATTTCCATTGCTT	CAGTGAAATAGCTGATCGATTAGAAG
*sarA*	TTTCTCTTTGTTTTCGCTGATGT	TGTTATCAATGGTCACTTATGCTG
*agrB*	ACAGTGAGGAGAGTGGTGTAA	AGCTAAGACCTGCATCCCTA
*rsh*	CGAAACCTAATAACGTATCAAATGC	TGTATGTAGATCGAAAACCATCACT
*cidA*	GATTGTACCGCTAACTTGGGT	GCGTAATTTCGGAAGCAACAT
*fnbpB*	AATTAAATCAGAGCCGCCAGT	AATGGTACCTTCTGCATGACC

## Data Availability

The raw data supporting the conclusions of this article will be made available by the authors, without undue reservation.

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
