# Peer review of "Staphylococcus aureus Behavior on Artificial Surfaces Mimicking Bone Environment"

_pathogens, 2023, doi:10.3390/pathogens12030384_

Round 1

Reviewer 1 Report

Review “Staphylococcus aureus behavior on artificial surfaces mimicking bone environment” submitted for publication in Pathogens

Major

Abstract difficult to follow (see examples below) and needs careful revision to improve clarity.

Line 171: As performed here the mRNA content of the cells will no longer reflect mRNA content in planktonic/biofilm cells but instead reflect changes in transcription induced by shifting cells to room-temperature/centrifugation  (cells for transcriptomic analysis were pelleted for 5 min at room temperature before freezing).  To avoid breakdown of mRNA by endogenous RNases (the average half life of mRNA is only a few minutes), and to prevent RNA-transcription to adjust to the new growth conditions (here room-temperature), it is crucial that cells intended for RNA purification are cooled down immediately or treated with RNA-stabilizing agents.

Line 208, to the best of my knowledge S. aureus do not have pili. The blue arrows seem to be pointing to DNA. Accordingly, many USA300 cells appear lyzed.

Line 228, I do not follow the analysis of the biofilm composition, it is written the WGA stains exopolysaccharides, but since WGA is known to bind peptidoglycan and teichoic acids is the cell wall of S. aureus, I guess the WGA staining simply binds to S. aureus cells not to a putative exopolysaccharide responsible for gluing bacteria together in a biofilm?

Minor

Line 22+23: unclear sentence How ca a matrix be more important and what is the link to the following sentence: “whereas no difference in biofilm genes expression was observed between the two tested 23 surfaces”.

Line 24: I do not think “propose” is the right word here. Do you mean develop?

Line 25-27: unclear sentence, please clarify

Line 118: Minimal medium?

Line 111-114. What is the rationale for using these thee 3 strains: JE2 and SH1000 both belong to CC8. SH1000 derived from a UV exposed lab strain to cure it from  prophages)

Line 228: unclear sentence “For USA300 strain, the relative quantity of biofilm matrix was inferior compared to the two other strains. What is meant here? That the strain makes less biofilm?

Author Response

Reviewer 1

Review “Staphylococcus aureus behavior on artificial surfaces mimicking bone environment” submitted for publication in Pathogens

We thank the reviewer for his/her comments that will for sure improve our manuscript.

Major

Abstract difficult to follow (see examples below) and needs careful revision to improve clarity.

We thank the reviewer for this comment and we modified the abstract thanks to the reviewer’s comments.

“Infections, which interfere with bone regeneration, maybe a critical issue to consider during the development of biomimetic material. Calcium phosphate (CaP) and type I collagen substrates, both suitable for bone-regeneration dedicated scaffolds, may favor bacterial adhesion. Staphylococcus aureus possesses adhesins that allow binding to CaP or collagen. After their adhesion, bacteria may develop structures highly tolerant to immune system attacks or antibiotic treatments: the biofilms. Thus, the choice of material used for scaffolds intended for bone sites is essential to provide devices with the ability to prevent bone and joint infections by limiting bacterial adhesion. In this study, we compared the adhesion of three different S. aureus strains (CIP 53.154, SH1000, and USA300) on collagen- and CaP-coating. Our objective was to evaluate the capacity of bacteria to adhere to these different bone-mimicking coated supports to better control the risk of infection. The three strains were able to adhere to CaP and collagen. The visible matrix components were more important on CaP- than on collagen-coating. However, this difference was not reflected in biofilm gene expression for which no change was observed between the two tested surfaces. Another objective was to evaluate these bone-mimicking coatings for the development of an in vitro model. Thus, CaP, collagen-coatings, and the titanium-mimicking prosthesis were simultaneously tested in the same bacterial culture. No significant differences were found compared to adhesion on surfaces independently tested. In conclusion, these coatings used as bone substitutes can easily be colonized by bacteria, especially CaP coating, and must be used with an addition of antimicrobial molecules or strategies to avoid bacterial biofilm development.”

Line 171: As performed here the mRNA content of the cells will no longer reflect mRNA content in planktonic/biofilm cells but instead reflect changes in transcription induced by shifting cells to room-temperature/centrifugation  (cells for transcriptomic analysis were pelleted for 5 min at room temperature before freezing).  To avoid breakdown of mRNA by endogenous RNases (the average half life of mRNA is only a few minutes), and to prevent RNA-transcription to adjust to the new growth conditions (here room-temperature), it is crucial that cells intended for RNA purification are cooled down immediately or treated with RNA-stabilizing agents.

We thank the reviewer for this very interesting information. Indeed, we rapidly centrifuge the samples to obtain a pellet and to freeze them in -80°C, with no time for new growth conditions. All samples were treated under the same conditions. We modified the text.

“Bacteria from planktonic growth or adherent were rapidly collected by centrifugation at 5000 rpm, 5 minutes, at room temperature. Pellets were directly stocked at -80°C for long-term conservation.”

Line 208, to the best of my knowledge S. aureus do not have pili. The blue arrows seem to be pointing to DNA. Accordingly, many USA300 cells appear lyzed.

We agree with the reviewer. Thus, we wrote “pili-like” as we cannot to be sure of the nature of the structure pointed by the blue arrow. USA cells were not detected as dead when marked with Syto9 and PI fluochromes. The structure observed could be extracellular secretion.

Line 228, I do not follow the analysis of the biofilm composition, it is written the WGA stains exopolysaccharides, but since WGA is known to bind peptidoglycan and teichoic acids is the cell wall of S. aureus, I guess the WGA staining simply binds to S. aureus cells not to a putative exopolysaccharide responsible for gluing bacteria together in a biofilm?

We agree with the reviewer’s comment. WGA is known to bind to saccharides and so to peptidoglycan. However, WGA was used by another lab to bind to exopolysaccharides (Arciola CR, Campoccia D, Ravaioli S, Montanaro L. Polysaccharide intercellular adhesin in biofilm: structural and regulatory aspects. Front Cell Infect Microbiol. 2015;5:7). We added the reference in the publication. Moreover, in 3D reconstruction, the signal of WGA was localized at different positions than bacteria, around, on, and between them.

Minor

Line 22+23: unclear sentence How ca a matrix be more important and what is the link to the following sentence: “whereas no difference in biofilm genes expression was observed between the two tested 23 surfaces”.

We modified this sentence.

Line 24: I do not think “propose” is the right word here. Do you mean develop?

The reviewer is right, we change by “evaluate”.

Line 25-27: unclear sentence, please clarify

We modified this sentence.

Line 118: Minimal medium?

We modified minimum in minimal medium.

Line 111-114. What is the rationale for using these thee 3 strains: JE2 and SH1000 both belong to CC8. SH1000 derived from a UV exposed lab strain to cure it from  prophages)

We add the information explaining this choice.

“Three strains of S. aureus were used in this study: a methicillin-resistant strain USA300 isolated in the 1990s in the USA and two methicillin-susceptible strains: CIP 53.154 (ATCC 9144) isolated in Oxford in 1944, used as referent strain in European Pharmacopeia and SH1000, derivate from the strain 8325-4, with the rsbU gene reconstituted, which forms strong biofilms. Trypto-Casein Soy Agar (TSA, Biokar, Allonne, France) was used for the isolation and cultivation of tested bacteria.”

Line 228: unclear sentence “For USA300 strain, the relative quantity of biofilm matrix was inferior compared to the two other strains. What is meant here? That the strain makes less biofilm?

Indeed, the USA strain formed less biofilm than the two others under the tested conditions and in our hands as discussed in the discussion part: “Biofilm matrix formed on this support was less important for USA strain but mainly composed of exopolysaccharides for all strains. However, the proportion of proteins was strain-dependent, confirming the previous description of S. aureus biofilm (22).”

Reviewer 2 Report

The research has been developed professionally, the methods used in the research are justified and the results obtained are convincing. However, the conclusions contain a presumption rather than based on the research results. It could be, with a high probability of using the antimicrobials to reduce bacterial adhesion, but the study had other aims - to study this adhesion rather than exposure to antibiotics. The conclusion should be formulated on the basis of the results.

Author Response

Reviewer 2

The research has been developed professionally, the methods used in the research are justified and the results obtained are convincing. However, the conclusions contain a presumption rather than based on the research results. It could be, with a high probability of using the antimicrobials to reduce bacterial adhesion, but the study had other aims - to study this adhesion rather than exposure to antibiotics. The conclusion should be formulated on the basis of the results.

We thank the reviewer for these comments that improve our manuscript. In consequence, we modified our conclusion to stay based on the described results.

“In conclusion, CaP-based bone substitutes allowed S. aureus adhesion and could enhance the risk of bone and joint infection. Collagen bone substitute seems to be less risky, but is still easily subject to bacteria adhesion, especially if collagen is used in 3D structure, which offers more surfaces and the possibility for bacteria to adhere.”
